# Use of instructional videos in leadership education in higher education under COVID-19: A qualitative study

**Daniel T. L. Shek** (ORCID)*, **Tingyin Wong** (ORCID), **Xiang Li, Lu Yu**

Department of Applied Social Sciences, The Hong Kong Polytechnic University, Hung Hom, Kowloon, Hong Kong, China

* daniel.shek@polyu.edu.hk

## Abstract

The use of online teaching mode has grown rapidly in recent years, particularly under the COVID-19 pandemic. To promote the learning motivation of students and teaching effectiveness, development of attractive online teaching material such as videos is important. In the present study, we developed 15 theory-related videos and 9 case-based videos in the context of a leadership course focusing on psychological well-being and psychosocial competence. Using a qualitative research methodology via focus groups (N = 48 students) to evaluate these videos, six themes emerged from the data, including video arrangement, design of videos, content of videos, benefits to students' pre-lesson self-learning, benefits to students' learning of course content, and contribution to students' class participation. The findings suggest that the videos can elicit positive perceptions of the students in a flipped classroom arrangement. Students also benefit from the videos in terms of their understanding of course content and their participation in class discussion. Besides, the study suggests that the videos promote the learning efficiency of the students. The present qualitative findings concurred with the previous quantitative findings, suggesting the value of using virtual teaching and learning to promote psychosocial competence in university students.

## 1. Introduction

The use of online teaching mode in higher education institutions has grown rapidly in recent years. Before the COVID-19 pandemic, more than 5.5 million students in the U.S. were enrolled in one or more postsecondary-level courses through online channels [1]. Researchers highlighted that studies on video-based learning has increased significantly after 2006 because of the increased use of videos in higher education sector, both as a supplementary materials for face-to-face lectures or as the main way to deliver online education content [2, 3]. During the early phrase of the pandemic, videos played a key role in sustaining teaching and learning in higher education [4]. For example, in Beijing Normal University, 3238 out of the 4036 courses in February 2020 were offered online [5]. With the increased use of digital device and various online pedagogies (e.g., videos and virtual reality), students may benefit from flexible online

Professor of Applied Social Sciences OR Professor Elsie Yan (elsie.yan@polyu.edu.hk), Chairperson of the Departmental Research Committee overseeing research ethics approval, The Hong Kong Polytechnic University. The data are not publicly available due to serious ethical concerns regarding the publication of focus group interview transcripts in a public repository. Specifically, when the data were collected, participants only gave consent for the findings to be published anonymously for educational and research purposes. Making the data public would violate this consent.

**Funding:** This project was financially supported by the University Grants Committee on the development of virtual teaching and learning materials (3.54.xx.89PZ), and Dr. Daniel Shek Matching Fund from PolyU (1.54.xx.52UK). The preparation of this paper was supported by the Li and Fung Endowed Professorship for Dr. Daniel Shek at The Hong Kong Polytechnic Universit.

**Competing interests:** The authors have declared that no competing interests exist.

teaching mode and technology-supported active learning. Moreover, researchers argued that the post-pandemic pedagogy could harmoniously integrate technology and physical tools because of technology's high penetration among Generation Z students, its capability in enhancing teaching and learning effectiveness, and its potential of preparing subjects for unknown future challenges [6, 7].

Online education could be delivered through synchronous (i.e., real-time lecture through the internet) and asynchronous (i.e., pre-recorded lectures) teaching and learning modes. Also, blended or hybrid learning (i.e., combination of online education and traditional classroom teaching) has been shown to optimize students' online learning experience under the pandemic [8, 9]. Additionally, massive open online courses (MOOCs), which are non-credit online courses for learners not enrolled in higher education institutions, were offered by private companies (e.g., Udacity and Coursera) and universities (e.g., Stanford University) to millions of learners worldwide [10].

Video-based learning is defined as the learning through the use of videos, which is comprised of visual and audio cues to present information for learners [11]. It is recognized as a powerful resource for online teaching and learning [12]. Under the pandemic, the use of video-based learning in online education also increased. It was found that, when the lockdown measures were in place during the pandemic, there was a significant increase in the number and diversity of didactic resources (i.e., video and audio files) in flipped classroom sessions [13]. Video-based learning could bring various learning benefits to students, including enhancing learning outcome, providing flexibility to learn at their own pace, and improving participation in online discussion [11]. In addition, videos could be combined with various teaching methods, including mobile-assisted learning and flipped classroom approach, to optimize students' learning process. For example, exploring the use of videos and texts in mobile learning, Reychav and Wu [14] and Wang [15] found that videos could improve students' individual learning outcome. Combining with the flipped classroom approach, Wikandari et al. [16] found that animated videos could significantly enhance university students' ability in comprehension and application. Obviously, there is a need to ask whether virtual teaching and learning tools (e.g., videos) can promote the mental health of students, which is an important pillar in public health.

Under the pandemic, there was an increased use of emergency remote teaching because school lockdown policies were implemented in many countries [17–20]. However, in online learning, students may feel isolated, bored, anxious or frustrated. The lack of instant feedback from teachers may decrease their motivation [21–23]. In hybrid learning, students may face difficulties in self-regulation [24]. Moreover, online learning may pose public health risk for adolescents. Studies suggested that prolonged exposure to digital devices may lead to mental health problems in learners. Stressors related to quarantine and lockdown plus mental health problems from prolonged screen time may result in burnout and exhaustion in learners [25]. Therefore, it is important to design curriculum materials that could motivate students to learn and reduce their risk of developing mental health problems in an online learning environment. Studies showed that instructional videos could enhance students' interest in the course content and their learning motivation [26, 27]. Based on the cognitive theory of multimedia learning, Chen [28] found that videos provided self-controlled learning environment for students, so they could easily locate the content they want. The concise text and images in videos could also reinforce students' learning. Moreover, the audio and visual stimuli in videos could enhance learners' learning effectiveness and reduce their extraneous cognitive load when processing information. Video was also found to be a more effective learning tool than the sole presentation of text [29–31]. In the post-pandemic period, researchers viewed instructional videos as valuable original online learning resources for higher education courses [32].

Since 2012/13 academic year, we have implemented a leadership course entitled "Tomorrow's Leaders" for undergraduate students at a Hong Kong university. The positive youth development (PYD) approach was used in the development of this course, which emphasizes fostering of adolescents' cognitive, emotional and social competences to ensure successful childhood-to-adulthood advancement [33, 34]. The "Tomorrow's Leaders" course aimed at enhancing university students' intra- and interpersonal competences to equip them to meet the changing needs for success in the 21st century [35]. Moreover, this course was intended to promote university students' leadership qualities, so they could be leaders for themselves (i.e., self-leadership) and lead others within their social circles [36]. In the "Tomorrow's Leaders" course, we have developed videos based on various PYD attributes, including self-leadership, cognitive competence, social-emotional competence, resilience, moral competence, spirituality, law-abiding leadership, cultural competence, effective communication, and team building. Empirical studies argued that leadership qualities, including self-leadership, psychosocial competencies, interpersonal relationships, problem solving, conflict resolution, and stress management, played a key role in strengthening children's and adolescents' mental health [37–39]. Besides, some leadership qualities such as resilience and psychosocial competence can be regarded as integral components of mental health. Because of the close connection between adolescents' leadership qualities and their mental health, this study could provide implications for public health. Our previous evaluation study of this course showed that students had significant positive changes in a number of PYD attributes after taking the course [40], and high satisfaction with the course content, course design, teaching methods, and teaching and learning effectiveness [41, 42]. Under the pandemic, the delivery of this course was changed from face-to-face mode to online teaching or HyFlex learning (i.e., a flexible combination of online and face-to-face teaching) [43, 44]. In view of the benefits of instructional videos, a research project was implemented to develop virtual teaching and learning materials for the "Tomorrow's Leaders" course, with 15 theory-related videos and 9 scenario-based videos produced to facilitate students' learning in this course. The details of the videos can be seen in Shek et al. [45].

After the development of the 15 theory-related videos and the 9 scenario-based videos, we evaluated the quality of the videos by employing the subjective outcome evaluation method via the client satisfaction approach. Our first study was conducted in Semester 2 of 2021/22 academic year, in which the course was delivered in an online synchronous mode. Two subjective outcome evaluation scales (SOES) were developed to assess students' views of the theory-related videos and the scenario-based videos, respectively. More than 90% of the students perceived that the videos could promote their interest, understanding and reflection of the subject matter, and were beneficial to their development [46]. Furthermore, we replicated the study in Semester 1 of 2022/23 academic year in order to analyze students' perceptions of the 24 videos under face-to-face teaching mode. Using the same measures, the second study found that students' evaluation of the video quality and benefits received from watching the videos were significantly improved [47].

Although the two subjective outcome evaluation studies showed that students had highly positive appraisal of the 24 videos, there was a need to use qualitative evaluation strategies to further understand students' perceptions of and experience in watching the videos in this course. Qualitative research examines the nature of human experience and the meaning of certain phenomena to individuals, so that individuals' interpretation and explanation of the "what", "how" and "why" of a phenomenon could be explored [48]. For education research, it was stressed that qualitative research could comprehensively explain human beings' psychological dimensions and the nature of their complex behaviors [49]. Moreover, focus group interview is an effective qualitative method to examine a particular issue in which a group of participants have experienced together. It is suitable when researchers seek to obtain a range of

ideas or feelings from participants about certain issue, or examine the factors that affect individuals' viewpoints or behavior [50]. Previous studies have used focus group studies to evaluate the impacts of videos on students' online learning [51, 52].

### 1.1. Research questions

In two previous quantitative studies, we examined students' perceptions of the theory-related and case-based videos' content, implementation quality and effectiveness [43, 44]. Generally speaking, the findings of these two studies are very positive. To understand the experience of the students, we conducted the present qualitative study using the focus group method to answer the following three research questions:

1. What are students' perceptions of the pre-lesson video-watching arrangement?

2. What are students' perceptions of the quality (i.e., video content and video design) of the theory-related videos and case/scenario-based videos?

3. What are students' perceptions of the benefits gained from watching the theory-related videos and case/scenario-based videos?

## 2. Materials and methods

The present study used a focus group method to evaluate the perceived quality and effectiveness of 24 instructional videos in a leadership course in a Hong Kong university. Data were collected in Semester 2 of 2021/22 academic year, during which students attended the lectures through an online synchronous teaching mode. Students were asked to watch two to three theory-related videos and scenario-based videos before attending the 2-hour online synchronous lecture (i.e., flipped classroom approach) every week. To gain participation points, students needed to complete the blended learning exercise and have reflections related to the videos before lecture each week. In total, 10 lectures that taught different leadership qualities, including intrapersonal development (i.e., mental health) and interpersonal development, applied this video watching arrangement. Focus group interviews were conducted at the end of the semester when students had completed all the lectures in the course.

### 2.1. Research design

In terms of the philosophical orientation of this study, a general qualitative research orientation was employed [53, 54]. The general elements of qualitative studies emphasize naturalistic inquiry, personal experience of research participants, empathic neutrality of the researchers, and an inductive analysis approach. Additionally, qualitative studies could examine individual experience or describe common meaning of several individuals' lived experiences to develop composite description of the essence of participants' experience [55, 56]. In this study, we focused on developing patterns of meaning from students' subjective understanding of their experience. The design of interview questions was mainly broad and general, so students were able to express their own understanding of an experience. Besides, focus group interviews allowed meaning to be constructed through students' interactions with others. We also upheld principles proposed by Shek, Tang and Han [54], including highlighting the philosophical orientation of the study, detailed description of different parameters of the study, building up consensus and triangulation of findings based on different methods.

## 2.2. Participants and procedure

In Semester 2 of the 2021/22 academic year, the "Tomorrow's Leaders" course was delivered to 1,308 Year 1 undergraduate students at a Hong Kong university. These students majored in different subjects under seven schools or faculties at the university, including School of Design, School of Fashion and Textiles, School of Hotel and Tourism Management, Faculty of Construction and Environment, Faculty of Engineering, Faculty of Health and Social Sciences, and Faculty of Science. At the end of the semester, the teachers of the "Tomorrow's Leaders" course purposefully sampled students to participate in the focus group interviews. The sampled participants were homogeneous because all students were Year 1 students at the same university. This sampling strategy fits the research purpose because the study aimed at understanding the common experience of students rather than comparing the difference between them. Financial incentive was also given to encourage students' participation. At the end, 53 students agreed to join the interviews with 48 of them successfully participated in a voluntary manner.

In the current study, research participants' anonymity was maintained, meaning that students' demographic information was not collected. Maintaining research participants' anonymity would enable them to disclose more information or opinion during the interview [57, 58]. The sampled participants were all year 1 students in the case university. The general student demographic information in the case university showed that the mean age of year 1 students in the university is 18 and there is an equal ratio of male to female students [59]. Analyzing the demographics of research participants is important to understand whether findings of the current research could transfer to other settings or contexts [60, 61]. These demographic characteristics were generally similar to students in other Chinese universities [62] and international universities [63]. Therefore, the findings of the current study could be generalized to students in Chinese or international contexts. However, it is important to note that students' perceived benefits from watching educational videos were influenced by various factors, including the subject taught and external social factors (e.g., COVID-19 pandemic) [64–66]. Moreover, studies also noted that students' financial status could affect their ability to acquire electronic devices, thus influencing the effectiveness of electronic learning [67]. Readers should consider these factors when applying the current research findings in other contexts.

Because of social distancing measures, all focus group interviews were conducted online. Before conducting each interview, the interview protocol, data confidentiality, students' voluntary participation and the protection of their privacy were explained to students. Besides, students' consent to the recording of the interviews was sought. The 48 students were conveniently divided into 8 groups, ranging from 3 to 9 students per group. The first focus group interview was conducted on 23 May 2022, while the last interview was completed on 17 June 2022. The interview length ranged from 1 hour 34 minutes to 3 hour 1 minute, with the first interview conducted in English and the rest delivered in Chinese. Table 1 lists the number

**Table 1. Duration of the interview, number of participants, and language used in each focus group interview.**

| Focus Group Number | Group 1 | Group 2 | Group 3 | Group 4 |
|---|---|---|---|---|
| Duration of the interview | 1:43:25 | 2:51:35 | 2:00:22 | 3:01:18 |
| Number of Participants | 7 | 9 | 6 | 8 |
| Language Used | English | Chinese | Chinese | Chinese |
| Focus Group Number | Group 5 | Group 6 | Group 7 | Group 8 |
| Duration of the interview | 2:31:18 | 2:37:42 | 1:34:45 | 1:44:08 |
| Number of Participants | 5 | 6 | 3 | 4 |
| Language Used | Chinese | Chinese | Chinese | Chinese |

of participants, the duration, and the language used in each focus group interview. Besides, each focus group interview had two moderators, with the first moderator hosting the interview and the second moderator asking follow-up questions and taking notes. All moderators of the interviews have rich experience in qualitative research.

## 2.3. Instruments

Qualitative research focuses on investigating "what" the participants have gone through and "how" they have experienced it [53, 56]. Following this principle, a semi-structured interview guide was developed for the focus group interview (Table 2). The interview guide was divided into five sections: 1) arrangement of watching these videos, 2) general learning experiences, 3) theory-related videos, 4) scenario videos, and 5) conclusion. Each section contains three to seven interview questions. In semi-structured interviews, the use of probing strategy and more open-ended questions by interviewers enable the understanding of more complex issues [68]. Moreover, open-ended questions are preferred in qualitative research because they allow participants to provide more description of their experience, which could lead to richer textual and structural description of participants' experience in the final report [56].

## 2.4. Data analyses

According to Polkinghorne [69], the accuracy of transcription is one of the criteria for judging the quality of a qualitative study. Therefore, the 8 focus group interviews were verbatim transcribed, and then double-checked by an experienced researcher. Qualitative research does not only describe what participants have experienced, but its data analysis focuses heavily on researchers' interpretation [56]. Therefore, the transcripts were coded using line-by-line coding, which refers to the coding of each line of text in the transcripts. It enables researcher to develop codes that reflect the experience of interviewees rather than the theoretical beliefs of the researcher [70].

To ensure that the data analysis was systematic and verifiable, line-by-line coding followed by constant comparison of the codes between focus groups were conducted [50]. The line-by-line coding identified 92 first order codes. At the secondary coding stage, insignificant codes were disregarded and similar codes were grouped under 22 second order codes, which could be combined into six themes. Besides, the frequency of each first order code was calculated. Sim [71] noted that group dynamics of each focus group would affect the occurrence of the first order codes. However, comparing the difference in topics discussed in each group, only a few topics were more frequently deliberated in some groups, which did not significantly affect the interpretation of the interview data. So, the frequencies of the first order codes could serve as a reference to the strength of participants' opinions.

Validity in qualitative research determines the accuracy, trustworthiness and credibility of the account or perceived realities of the researchers and research participants [72]. It focuses on the researchers' care in the analysis procedures. Reliability assesses the agreement on codes by multiple coders. Researchers stress that qualitative research emphasizes more on validity, while reliability only plays a minor role [73]. The concept of trustworthiness, containing four aspects (credibility, transferability, dependability and confirmability), was also used by researchers to understand validity and reliability in qualitative research [72]. Additionally, researchers supported the importance of applying various methodological strategies to demonstrate the qualitative rigor [74]. Methodological strategies commonly applied in qualitative researchers include audit trail, member checking, peer debriefing, and rich description [74, 75]. In the current study, different strategies were applied in the research design and methodology to ensure that findings of the research were valid and reliable. First, audio recording the

**Table 2. Semi-structured interview guide.**

| |
|---|
| **1. The arrangement of watching these videos** |
| 1.1 Have you encountered any difficulties in watching these video clips? If yes, what are they? How do you overcome these difficulties? Can you provide some examples? |
| 1.2 How much time did you spend in watching the video clips on average every week? Usually, how many times did you watch the video clips? |
| 1.2.1 Do you watch them before or after the on-site class? If you like some videos, did you watch them again after attending the lecture? |
| 1.2.2 Did your teacher play the video again in the face-to-face/online class? If yes, any benefits for your learning? Do you enjoy watching these videos again? |
| 1.3 How do you perceive this arrangement (i.e., you are expected to watch the videos before attending the lecture)? What are the benefits and shortcomings of watching these video clips before lectures? Does watching the videos increase your study load? Do you enjoy watching these videos? Any suggestions on this arrangement? |
| 1.4 Can watching the videos reinforce your participation in the face-to-face/online real-time class and facilitate your understanding of teachers' instruction? |
| **2. General learning experiences** |
| 2.1 These video clips aim to provide some background information to facilitate your understanding of related theories. Do you think the video clips can facilitate your understanding of the theories? |
| 2.1.1 If yes, how do the video clips help you understand the theories and support you to become more engaged in the class? |
| 2.1.2 If not, why not? How to improve (e.g., the quality of the videos, the use of the videos)? Any suggestions? |
| 2.2 In general, what do you think about the quality of the video clips (e.g., content, audio, subtitles, number of video clips, duration of the video clips, digital effects)? |
| 2.3 For each lecture (i.e., Lecture 2 to 11), we prepare one or more than one theory-related videos and case scenario videos. Do you think the theory-related videos and the scenario videos are closely related? |
| 2.3.1 If yes, any impressive examples/cases among the 10 lectures? |
| 2.3.2 If not, how to strengthen the links between the two kinds of videos? Or which kind of videos can be removed? |
| **3. Theory-related videos** |
| 3.1 Do you think the content of the theory-related videos is clear enough for self-learning? Is it difficult for your self-learning? |
| 3.2 Do you think the theory-related videos are closely related to the relevant lecture? Can the theory-related videos help you better understand the theories? Any impressive examples/cases among the 10 lectures? |
| 3.3 What aspects/components of the theory-related videos can benefit your learning (e.g., model explanations)? |
| 3.4 Are these theory-related videos helpful to your learning? Can you recall the two most impressive videos among the 15 theory-related videos? Why do you like these two videos most? What are the strengths of these two videos? |
| 3.5 Please share one or two theory-related videos that need to be revised/improved. Why are they not helpful for you? How could we improve them? |
| **4. Scenario videos** |
| 4.1 Do you think the content of the scenario videos is clear enough for self-learning? |
| 4.2 Do you think that the scenario videos are closely related to the relevant lecture? Can the scenario videos help you better apply the theory to your daily life? Any impressive examples/cases among the 10 lectures? |
| 4.3 What aspects/components in the scenario videos can benefit your learning (e.g., apply the theory in practice, very related to my life, very practical)? |
| 4.4 Are these scenario videos helpful to your learning? Can you recall the two most impressive videos among the 9 scenario videos? Why do you like these two videos most? What are the strengths of these two videos? |
| 4.5 Please share one or two scenario videos that need to be revised/improved. Why are they not helpful for you? How could we improve them? |
| **5. Conclusion** |
| 5.1 Overall speaking, are you satisfied with the video clips used in this course? If the full mark is 10, ranging from 1 to 10, how much will you give? |
| 5.2 To benefit your learning and strengthen your understanding of specific topics, do you have any suggestions to improve the video clips? |
| 5.3 What else would you like to share? |

interviews allowed researchers to repeatedly check whether the emerging themes truly represented participants' accounts [61]. Second, rich verbatim extracts enabled readers to judge the truthfulness of the final themes [75, 76]. Both of these strategies were used in this study to ensure that the findings represented the data collected, thus enhancing the validity of the research. Third, emerging themes were discussed transparently among the researchers in this project to ensure that biases in the research method were minimized, which could enhance the reliability of this study [61, 72]. Fourth, inter-rater reliability was conducted to verify the strength of the findings [54, 77]. 20 randomly selected narratives from the focus group interviews were presented to two independent researchers and they were asked to re-code these narratives. The result showed that the two researchers agreed on 18 out of 20 codes. This showed a high level of inter-rater reliability in the analyses (inter-rater reliability = 18/20 = 0.9). These two strategies ensured that researchers arrived at similar codes or themes in the data analysis, thus enhancing the reliability of the current research. Fifth, triangulation was a typical strategy to enhance the validity of research findings [73, 75]. The triangulation of the findings in the current study and findings from our previous quantitative study is presented in the discussion section of this paper.

## 3. Results

This section presents and analyzes the first and second order codes under six categories of responses (i.e., themes). The six themes are: 1) video arrangement, 2) video design, 3) video content, 4) videos' benefits to the students' pre-lesson self-learning, 5) videos' benefits to the students' learning of course content, and 6) videos' benefits to the students' class participation. These themes are presented below.

### 3.1. Video arrangement

In terms of the video arrangement, this study examined three aspects: 1) positive effect on students' learning, 2) connection between the videos and the lectures, and 3) drawback to students' learning.

First, for the positive effect of the video arrangement, students expressed that the pre-lesson video watching arrangement allowed students to pace their learning according to their own needs and gain more flexibility (N = 6). With the addition of the videos and the blended learning exercise, lecture time was reduced from 3 hours to 2 hours, which enhanced their concentration in lectures (N = 9). As elaborated by a student:

> "This course used to have 3 hours lectures. However, with the addition of the e-learning modules, including watching the videos and answering [the blended learning] questions before lectures, lecture time is reduced to 2 hours. This is beneficial to my concentration in lectures. My concentration is better in 2-hour lectures. Without these videos, we would have 3-hour lectures. Even though we could take a break in the 3-hour lecture, the learning outcome would not be very good." (Group 8 Line 68–70)

This arrangement also allowed students to gain better understanding of the lecture content and engage more in class discussion (N = 19). As stated by one student:

> "I think the arrangement is reasonable. The videos are like pre-lesson preparation materials. There are many theories introduced in the videos. If we can't digest it in class, we can't discuss it on the spot. Watching these videos before class gives us time to digest, so we can better answer questions about this topic in class discussion." (Group 6 Line 130–132)

Watching the theory-related videos before class allowed more time for teachers to share personal experience or conduct class discussion during lectures, which was more interesting and beneficial to some students (N = 2). Besides, many of the students thought that the arrangement of video watching and blended learning exercise was fair because it counted towards their participation grade (N = 18), which forced them to do the exercise, reflect on their daily issues, and relate the theories to the cases (N = 6).

Second, in terms of the connection between the theory-related videos and the lectures, students believed that they were closely related because the teachers recapped the theories (N = 16), or discussed related personal experience and real life examples during lectures (N = 21). As elaborated by a student:

> "I think the teacher's explanations in the lectures and the theory-related videos are very relevant, because the teacher would use some daily examples to explain the theories, so they would not be so difficult to understand. For example, there is a more abstract topic called spirituality, which I think is difficult to understand. However, after we have self-studied the videos, the teacher would further explain how we could exercise this ability in our daily lives, such as cultivating more interests. It is related to what is mentioned in the videos. The videos also mention how we could exercise this ability, and there are some examples to explain it." (Group 6 Line 357–361)

Besides, in terms of the connection between the case-based videos and the lectures, the students thought that they were closely related because some teachers discussed the cases during lectures, while other teachers shared personal experience that were related to the videos (N = 20). Sometimes, teachers would share students' blended learning answers, or explain difficult concepts in the videos (N = 16). As one student remarked:

> "At the end of the [case-based] videos, there are some blended learning questions which require us to express our opinions on the online learning platform. Our teacher did not only share the views of the students, but also shared his own views. This has triggered my curiosity to read other students' responses [on the online learning platform]. So I think this motivates my study, and encourages my participation in class." (Group 7 Line 123–125)

These class discussions engaged students and increased students' understanding of the lecture topics. Teachers' interpretation of the videos and the extra examples offered other perspectives to students and enhanced their understanding.

Third, the video arrangement has some drawbacks. Some students thought that the arrangement exceeded their expectation (N = 1) or the workload was too big when they had midterm exams or assignments for other courses (N = 11). It was also stressful for them to understand so many concepts before attending lectures. Some students preferred watching the videos and completing the blended learning exercise after lectures because they would have better understanding of the topics after teachers' explanation (N = 4). As pointed out by a student:

> "I think it would be better for us to watch the videos and finish the related tasks after lecture. Before lecture, many students would forget to watch the videos. After lecture, the teacher has explained the concepts and elaborated using his/her life experience, so we would have more understanding of the concepts. When we watch the videos after lecture, it can deepen our understanding of the concepts. So I think it would be better for us to watch the videos and complete related tasks after lecture." (Group 5 Line131-133)

Students also expressed that they were not able to ask teachers questions immediately after watching the videos. In addition, some students stated that they may not pay full attention in lectures after understanding the course content from the videos before lectures (N = 2).

## 3.2. Video design

In terms of video design, this study investigated two aspects: 1) video quality and features that contributed to students' learning, and 2) general suggestions for improvement.

First, the majority of the students thought that the quality of the videos was good (N = 21). Students thought that the combination of the theory-related videos and the case-based videos provided a variety of video format, which could enhance their interest in the videos (N = 3). For case-based videos, students believed that the videos with real persons acting out the scenario (i.e., role-play videos) had good production. The topics of these videos were also related to students' lives (N = 3). As elaborated by a student:

> "I also think the quality is good. At least, the production has thought about the things that are closely related to [students'] daily life, especially those about the scenario-based videos. Some of the scenario-based videos are about group projects, some are about arguments between family members, etc. These topics are closely related to our daily lives." (Group 2 Line 379–381)

For theory-related videos, students perceived that various features of theory-related videos had contributed to their learning. Students enjoyed the animation (N = 3), and the interesting animation helped their understanding (N = 10). Besides, the theory-related videos were very structured and explained the theories clearly. Model explanation was helpful to students' learning (N = 16). The arrows, list and organization in the theory-related videos made the theories and the causal relationships easy to understand (N = 7). As stated by a student:

> "I think there are some definitions and models in the clip, which can help me understand the topic better. The animation and narration are very clear . . . step-by-step . . . what the first step is, and what the second step is. The videos let me clearly know how to proceed, the sequence and the like. They allow me to understand the topics more clearly, with guidance, and in a systematic way." (Group 2 Line 532–535)

Besides, the combination of visual and audio stimulation could improve students' concentration (N = 12). The analysis and discussion of examples in the theory-related videos and the blended learning exercise facilitated students' self-reflection and the development of critical thinking skills (N = 6). In the theory-related videos, the use of discussion or interactive mode for theory learning was more interesting for students (N = 3). However, in some of the theory-related videos, the script of the narration directly replicated the visual content of the videos, and students thought that this presentation style was boring (N = 3).

Second, in terms of ways to improve the video quality, students suggested improving the audio quality of the videos (N = 10). Moreover, students believed that the length of each video should be shortened (e.g. within 5 or 10 minutes) (N = 3), and they preferred the videos to have a faster pace (N = 2). For the case-based videos, students would like to have more variation of video format, so they could have stronger impression of the videos. Besides, students preferred story telling or role-play video production because these presentation styles were more interesting and engaging for them (N = 14).

### 3.3. Video content

In terms of video content, this study investigated three aspects: 1) theory-related videos' content, 2) case-based videos' content, 3) connection between theory-related videos and case-based videos, and 4) general suggestions for improvement.

First, in terms of theory-related videos' content, students encountered some difficulties. A number of students expressed that the difficult technical terms or English vocabularies in the videos hindered their understanding of video content (N = 15). As elaborated by one student:

> "My opinion is the same as that of the other students. First, the quality of the videos is very good, but what needs to be improved is that many videos have many definitions from different scholars. It would be better to add some daily examples. We checked the definitions on Google ourselves, and they are from different scholars. We can find all the definitions, but it is the definitions that bothered us because the wordings of the definitions are mostly vague and abstract. If the videos could use more conversational English and daily life examples, it will be easier for us to understand." (Group 6 Line 32–36)

Besides, students believed that some theory-related video content was too abstract (N = 3), and suggested adding more discussion of daily life examples in the theory-related videos (N = 2). Some students found that there were too many concepts in the theory-related videos, and recommended adding some guiding questions to help students' self-reflection. Also, students wanted to know the importance of each lecture topic to them personally and how the positive youth development (PYD) attributes would affect their lives.

Second, for case-based videos, various video features contributed to students' learning. The case-based videos were related to students' daily lives, so students could understand better and enjoy the videos (N = 7). As elaborated by a student:

> "I think the advantages of the case-based videos are that they are quite interesting, and they use more realistic situations. And another advantage is that I could feel the diligent production and a lot of effort was used, so I take the video watching very seriously." (Group 7 Line 367–368)

Moreover, students could relate the cases to social issues. The case-based videos provided various perspectives and students could learn the ways they could solve similar problems in the future (N = 14). However, students thought that there should be more case-based videos to explain more complicated theories or offer more elaboration and application of the theories in the scenario videos (N = 6). It was also believed that some case-based videos did not provide solutions and did not analyze different factors, which decreased the videos' benefits to students' learning (N = 3). Besides, some stories in case-based videos were too extreme (N = 3). Students also wanted the case-based videos to highlight the key points and provide some guided reflection on the topic. For cases that needed discussion, students wanted to watch cases that did not have conclusions, so there would be more room for discussion.

Third, this study investigated students' perceptions of the connection between the theory-related videos and case-based videos. The majority of the students agreed that the theory-related videos were related to the case-based videos (N = 24). The cases could explain the theories, and students could apply the theories into the cases. Moreover, the blended learning questions required students to use the theories to answer questions related to the scenarios, which further strengthened the connection between the two types of videos (N = 3). As elaborated by one student:

"I think those scenarios could echo the theories. One of the students mentioned just now the video about whether the husband should steal medicine for his wife's illness. In fact, the [blended learning] questions after the videos asked us to use the theory to answer. When I answer the questions, I would say from different levels whether the husband's behavior is in line with. . .that is, it may be unreasonable on the legal level, but if it is on a higher level of morality, it may be more reasonable. That is, we can use the theory to analyze the case, instead of only using general terms to discuss the matter." (Group 4 Line 604–609)

However, a few students thought that the connection between these two kinds of videos was not strong enough (N = 7). Besides, students suggested adding more real life application of the theories (N = 1) or simplifying the theory-related videos (N = 3) in order to strengthen the link between the theory-related videos and the case-based videos.

Fourth, for ways to improve the general video quality, students suggested adding translation of English vocabularies in the subtitles (N = 1). Moreover, students thought that the format or topics of some videos were quite outdated, so they suggested adding examples that were more related to the recent trend or news (N = 3) or discuss more social issues in order to strengthen their understanding of related topics (N = 1). The blended learning questions could be revised to strengthen their connection with the videos (N = 2).

## 3.4. Videos' benefits to the students' pre-lesson self-learning

In terms of videos' benefits to the students' pre-lesson self-learning, this study investigated two aspects: 1) theory-related videos' contribution, and 2) case-based videos' benefits.

First, the majority of the students thought that the theory-related videos were good enough for pre-lesson self-learning (N = 28) because there were clear explanation of theories and the examples in the videos could explain the application of theories. As explained by a student:

"I think the theory-related videos are easy to understand and suitable for self-study. Although sometimes I don't understand the content after watching the video once, I would get the concept after watching it two to three times, and the explanation is very clear. There are also examples, which can help my understanding. The theories are simplified for us, and the PowerPoint files summarized the theories. Compared to reading academic journals or only attending the lectures, these videos are definitely better." (Group 2 Line 466–469)

Also, students perceived the theory-related videos as effective learning tool and they were helpful for their term paper writing (N = 8). Besides, with the videos, students were able to pace learning according to their own needs and gain flexibility (N = 6). As elaborated by a student:

"I think the theory-related videos are easy to understand, and I didn't encounter any diffi-culties during the learning process. Especially during self-study, we can choose different ways to watch the videos. We can choose to watch the same video again. We can fast pace it, or we can pause it for a while at some point. Because videos have such flexibility, I think they are suitable for self-study." (Group 8 Line 211–213)

Besides, the theory-related videos could provide an overall mind map or guidelines (N = 5), and the pointers, highlighter and diagrams could explain the causal relationship in the theo-ries, which were helpful to students' self-learning (N = 4). However, some students thought that the theory-related videos were not clear enough for self-learning because the English

vocabularies were difficult to understand (N = 7), while a student believed that the general duration of the theory-related videos was too long (N = 1).

Second, in terms of the case-based videos' contribution to students' pre-lesson self-learning, the majority of the students thought that the case-based videos were suitable for pre-lecture self-study (N = 34) because they were related to students' lives, which can stimulate their reflection. Besides, the case-based videos matched the theory-related videos and demonstrated the concepts. As stated by a student:

> "I think these case scenario videos are suitable for self-study. Compared with the theory-related videos, they are easier to understand, because they are not just explanations in words, but they are some daily life examples to explain the theories concretely. These are supplements based on the theories, which are more suitable for us in pre-lecture preparation." (Group 2 Line 635–637)

The videos also used a casual and friendly tone to present the stories, which could engage students. As elaborated by a student:

> "Most of the scenario videos are interesting to me. For example, in one video, a Youtuber walks in [the university campus] and interviews different people. It is quite fun. It uses a very casual and friendly way to present, so it is suitable for us to watch." (Group 7 Line 278–280)

However, a small portion of students (N = 5) thought that while scenario videos were strict forward and easy to understand, they cannot be used to explain complicated concepts and may not motivate students to reflect.

## 3.5. Videos' benefits to the students' learning of course content

In terms of videos' benefits to the students' learning of course content, this study investigated three aspects: 1) videos' general benefits, 2) theory-related videos' benefits, and 3) case-based videos' benefits.

First, the majority of the students thought that the videos could facilitate their understanding of lecture content (N = 27) and related theories (N = 26). A student explained as follows:

> "It is a good starting point to use short videos to preview these concepts. I think this course has a lot of concepts or a lot of theories, so these videos can give us a preliminary understanding of these theories, then we can have more in-depth elaboration and application of these theories in class. I think this step-by-step approach is pretty good." (Group 4 Line 425–428)

Moreover, the videos allowed students to learn the course content with a self-discipline attitude, learn to manage their time (N = 5), and pace their learning according to own needs. The videos also increased students' learning flexibility and helped their assignment and term paper writing (N = 10). Besides, the combination of the theory-related videos and the scenario-based videos enabled students to understand both the theories and their application to daily life (N = 3). As elaborated by a student:

> "I think the videos can improve my understanding of related theories. Each topic's videos are divided into theory and cases. I think the combination is very good. After watching the theory-related videos, I am already familiar with those theories. So, when I watch the

scenario-based videos, I know how to apply the theories to real life examples. In this way, I can have a deeper and better understanding." (Group 2 Line 302–307)

The majority of the students rated the videos 7 points (N = 10), 7 to 8 points (N = 7) or 8 points (N = 18) out of 10 points. This showed that students were satisfied with the quality and benefits received from watching the 24 videos.

Second, the theory-related videos contributed to students' learning through clear explanation of theories, which helped students to understand the content without prior knowledge. The animation, layout and video cutting also made the theory-related videos more interesting. For this topic, students in the focus group interviews were asked to give examples of videos that had significantly contributed to their learning. As elaborated by a student:

"I think the video on Richardson's Model I just mentioned is good. If we don't have this video, I will look at this model for two hours but don't know what it is explaining. I think it's not just about these two videos, but all the videos are very clear. Even about concepts or models I haven't seen before, I can understand them from the videos." (Group 8 Line 276–278)

Moreover, some theory-related videos' content was related to students' daily lives, so they could stimulate students' self-reflection. The skills introduced in the videos were useful to students. For example, one student was particularly impressed by the theory-related videos for Lecture 10 repair strategies and Lecture 5 resilience. As the student explained:

"I like the theory-related videos for Lecture 10 repair strategies and Lecture 5 resilience. Repair strategies are problems we often encounter, such as how to solve problems when we quarrel with friends, however, I never thought about it. After watching the video, I would think about whether I can solve the problem according to the theory. In addition, because I didn't have a deep understanding of resilience at first, after watching the videos, I would know more about resilience. The content made me understand how we can cultivate this ability." (Group 6 Line 428–431)

Besides, in theory-related videos, there were concrete examples to explain the theories and illustrate the strategies for tackling different problems.

Third, students further elaborated on the case-based videos' benefits to their learning. The case-based videos applied theories to some real-life cases or social issue, so students could understand the concepts very easily. For this topic, students in the focus group interviews were also asked to give examples of videos that had significantly contributed to their learning. As explained by one student,

"There are two types of case-based videos. As mentioned just now, the first one stimulates daily situations. The other one, just like the case-based video on resilience, it can strengthen the reliability of the model [or theory]. It is not only the model talking about the concept, but the case-based video presents an example of how the protective factor can help a person to be more resistant to stress. This is more convincing, which impresses me even more. For example, there is a case-based video on misinformation, which is about the MMR vaccine injection. That video can help me apply the model to daily news. I usually only watch the news, and that is it. However, from the video, I learn that we could apply some of the things we have learned. So, I could gain another perspective." (Group 2 Line 745–748)

Also, the case-based videos were related to students' daily lives, so they could learn application of the theories in their daily lives. They could have self-reflection after watching the case-based videos, which was helpful for them to gain more positive mindset. As shared by a student:

> "I think the case-based videos gave me examples for reference on how to deal with the same situation, and how to deal with some people. They allow me to think more, and I would be aware of similar problems in my daily life, such as interpersonal skills and handling some situations. I could look at it from a third-person perspective. I would be calmer and think about how I should deal with it, or how I could ask for help from someone. I will know how to deal with similar situation. Sometimes when things happen to me, I will be very confused. However, from a third-person perspective, I will know how to solve it more clearly. I will have a more diverse perspective. Some of the videos are interviews, and some of them are positive or negative. I could think about the issues from different angles." (Group 3 Line 345–350)

Additionally, the stories of some of the case-based videos were entertaining and surprising, so they could engage students to have reflection even after the lectures. Some of the case-based videos were produced by real-life filming, while others were produced by animation. Some students thought that the animation in case-based videos were interesting and some video formats (e.g., interview) were new to students. The background music also allowed students to gain knowledge in a relax manner.

## 3.6. Videos' benefits to the students' class participation

In this study, many students perceived that both the theory-related videos and case-based videos could enhance their class participation (N = 16). Because they already knew the course content from the videos, they had time to understand and think about the topics before lectures, then they could share more during class discussion. As stated by a student:

> "After gaining some impression of the theories before class, I would have my own understanding. During the lecture, the teacher would give us some new ideas, which can back up the class discussion. Then, I can share more at this point, which can enhance the class discussion." (Group 8 Line 95–96)

Moreover, after watching the videos, students were able to answer teachers' questions (N = 16). The videos could also be used for class discussion, which increased students' motivation (N = 7). Also, students were able to gain more interest in the lecture topics after knowing the content from the videos (N = 20). As discussed by a student:

> "For class participation, I also think it is helpful, because after watching the videos on the topic, I can have more interest, which will make me particularly interested in those lectures and want to go to class." (Group 2 Line 305–307)

However, some students thought that the videos did not affect their participation in class because they believed that class participation depended on teachers' leadership, lecture topics, class atmosphere, class activities, other students' participation, or the personality and interest of students (N = 8). Some students stated that when they already knew the course content from the videos, they would be less concentrated in class discussion (N = 5).

## 4. Discussion

There is a growing interest and efforts in developing instructional videos for higher education students, particularly with the influence of the pandemic. The current research project is innovative in a number of areas. First, as the videos were offered to a large number of students (more than 2,000 students annually), development of effective instructional videos could benefit a large group of students every year. Second, both theory-related videos and scenario-based videos were developed, which was not common in Chinese and Western contexts. The combination of flipped classroom approach and video-watching was also rare in the Chinese context. Third, the findings in this qualitative study complement the quantitative findings obtained in our previous two studies. The quantitative studies provided support for the benefits of the 24 videos [46, 47], while this qualitative study enabled us to further understand how the videos had generated benefits for students.

In this study, the major findings were divided into six themes. First, in terms of video arrangement, students believed that the pre-lesson video watching arrangement enabled them to learn more flexibly, pace learning according to their own needs, and concentrate more in lectures. These results support other studies, which revealed that pre-lesson video watching was associated with better learning enthusiasm and academic results in undergraduate students [78, 79]. Besides, studies found that lecture videos enabled students to control their learning pace, which enhanced their satisfaction with the lectures and improved their academic results [80, 81]. This study also found that the teachers would recap the videos or discuss the blended learning exercise during lectures, which further reinforced the connection between the videos and the lecture. Studies revealed that the implementation of quizzes under flipped classroom arrangement, which was similar to the blended learning exercise in this study, could act as incentives for students to complete pre-class self-learning [82, 83] and could improve the effectiveness of the flipped classroom arrangement. The current study contributed to the literature by demonstrating how pre-lesson videos could be connected to lectures through class discussion. On the other hand, the current study also found that some students preferred watching the videos and completing the blended learning exercises after lectures. In this way, they would have better understanding of the topics after teachers' explanation in lectures. Literature also compared the learning effectiveness of pre- and post-lecture learning activities. Analyzing the effectiveness of different teaching and learning activities in an undergraduate course, Inglesi-Lotz et al. [84] found that 16.44% of the students thought that the pre-lecture exercises enabled them to have better understanding of the lecturers, while 25.84% of the students perceived that completing the post-lecture exercises facilitated them to revisit the lecture content and gain more understanding. On the other hand, Liang [85] revealed that pre- and post-lecture quizzes enhanced students' learning through different ways. The pre-lecture quiz motivated students' to reflect on their prior knowledge and arose their interest in the course content, while the post-lecture quiz increased students' awareness of their learning process and strengthened their metacognitive activities. These studies showed that pre- and post-lecture learning activities could be implemented to serve different learning objectives.

Second, in terms of video design, this study found that various features of the theory-related videos had contributed to students' learning. Students stated that the arrows, list and organization in the theory-related videos made the theories easy to understand. It was also found that combining visual stimulation and voice-over narration could improve students' concentration in the videos. These findings add to the literature on the multimedia learning theories, which examined how different video features could contribute to effective learning [32, 86–88]. The current research findings supported the "signaling" and "advance organizer" concepts in

multimedia learning theories [86, 89], which stressed that highlighting key points and using flow charts or tables for presentations reduced learners' extraneous cognitive load, thus improved their learning efficiency. Also, the "modality" principle was supported, which stated that presenting visual illustrations with voiceover narration could improve learning efficiency [32, 90]. For the features of case-based videos, some students in this study favored animation in the case-based videos because they were interesting to them, while some students preferred the use of role-play video production because it was more engaging. Studies also revealed that voice recordings in animated videos could bring cases to life [91], while stories with real-life filming could add a sense of realism and transfer textbook knowledge to practical settings for students [92]. Moreover, this study found that students perceived that the combination of theory-related videos and case-based videos offered a variety of video format, which enhanced their interest. In the literature, studies examined college students' and teachers' preferences for different video presentation format. It was found that videos on hands-on demonstration were most welcomed by college teachers [93], while videos that demonstrated instructors' presence were helpful to university students' learning [32, 94]. The current study adds to the literature by illustrating how theory-related videos and case-based videos could engage students' interest.

Third, for video content, in this study, students expressed that the difficult English vocabularies in the theory-related videos may bring additional challenge for them. In the literature, the majority of studies on video-based teaching and learning were conducted in the Western context, so this study showed how language could create a barrier for learners in the Chinese context. The "subtitle" principle proposed by Mayer et al. [95] also highlighted that having video subtitles was beneficial for students with diverse cultural or language background. Besides, this study found that the use of more realistic situations in case-based videos could arouse students' interest. In the literature, a study revealed that cases could link content with action, and could relate students' learning with others' experience [96]. Another study examining the use of videos to simulate clinical scenarios in nursing education also found that video simulation could effectively enhance students' critical thinking skills and leadership competencies, suggesting the usefulness of videos in helping nursing students to transfer textbook knowledge to professional practice [97].

Fourth, this study examined how the videos benefited students' pre-lesson self-learning. For theory-related videos, students believed that the videos enabled them to learn the theories in a self-learning mode because of the clear explanation and examples presented in the videos. For scenario-based videos, most of the students believed that they were able to self-learn the videos before lectures because the cases were related to their daily lives. In the literature, a study showed that flipped classroom approach combined with video-based learning materials were key to students' self-learning [15]. In practical dental training, a study also found that both two-dimensional videos and 360˚ virtual reality training enabled students' easy access and portability during online education and were effective for students' self-learning [98]. The current research adds to the literature by explaining how theory-related videos and case-based videos could benefit students' pre-lesson self-learning.

Fifth, this study investigated the videos' benefits to students' learning of course content. The videos enhanced students' self-discipline attitude and time-management. Watching both theory-related videos and scenario-based videos allowed students to learn the theories and their application. Also, this study showed that both theory-related and case-based videos could engage students in self-reflection. In the literature, studies demonstrated that animated videos on microscopic mechanisms explained the dynamic biological knowledge more clearly [99], and the use of case-based videos enabled students to experience scenarios with real patients even in a classroom setting [100]. Studies also showed that videos, especially case-based videos,

could facilitate students' meaningful learning and enhance their higher order thinking skills [101, 102]. A study on college-level sustainability courses revealed that working on "real world problem" could strengthen students' learning attitude, including self-regulation and self-responsibility [103].

Sixth, for students' class participation, this study found that the videos enabled students to learn and digest course content before class, so they would gain more interest in the topics and could answer teachers' questions during lectures. Students also highlighted that using the videos to induce class discussion could motivate them to participate in the discussion. Investigating the combination of animated videos and flipped classroom approach in a college-level biochemistry course, Wikandari et al. [16] argued that videos enabled students to learn at their own pace before attending lectures, while the flipped classroom approach allowed the implementation of interactive in-class problem solving activities. Besides, Danker [104] supported that the combination of flipped classroom and inquiry-based activities in lectures could engage students, enhance their curiosity and improve their higher-order thinking skills.

The objective of the "Tomorrow's Leaders" course was to enhance college students' leadership qualities through strengthening their positive youth development (PYD) attributes, which are important qualities of positive mental health. We have developed videos based on various PYD attributes, including self-leadership, cognitive competence, social-emotional competence, resilience, moral competence, spirituality, law-abiding leadership, cultural competence, effective communication, and team building. Our previous quantitative studies showed that the 24 videos were able to improve college students' learning and implementation of the leadership concepts in their daily lives [46, 47]. The current study used the qualitative research approach through the student focus group interviews. The findings showed that students had positive perceptions of the video content, video design and video arrangement. The results also revealed the videos were able to strengthen their pre-lesson self-learning, learning of course content and class participation, which may eventually contribute to the psychosocial well-being of the students.

Studies have shown the positive relationship between adolescents' leadership qualities and their mental health [37–39], so the production of these videos has public health implications. Furthermore, studies showed that mental health problems during adolescence, including depression, anxiety, personality disorders, eating disorder and substance abuse, would persist into adulthood [105, 106]. Besides, young people with mental disorders were likely to fail at key life events that happened during adolescence, including academic pursuit, job-hunting, friendships and romantic relationships establishment [107]. These young people had greater risks of having low social and economic status during adulthood [108, 109]. From the public health perspective, early interventions of adolescents' mental health problems could prevent the progression of primary and comorbid mental disorders into adulthood [107]. Therefore, the current project on developing educational videos for positive youth development could enhance adolescents' mental health and have implications for public health.

## 4.1. Triangulation of research findings

Triangulation could take four forms: comparing different data sources, involving multiple investigators, testing competing hypotheses, or including results from different research methods [75, 110]. In this study, the triangulation of methods was applied. The goal of this form of triangulation analysis is to look for convergence of results from different methods. When the results corroborate with each other, the validity of the findings is established [111]. Researchers supported the use of triangulation of different methods, including quantitative and qualitative approaches, to strengthen the validity of a study [53]. However, the implementation of

triangulation or mixed method requires the epistemological clarity of the different research methods [112]. Our quantitative studies adopted the subjective evaluation strategy through the client satisfaction approach, while the present study employed the qualitative paradigm focusing on participants' subjective experience. Both types of research approaches aim at investigating participants' perceptions of their experience, so the findings of our quantitative and qualitative studies could be triangulated.

In the two quantitative studies we conducted previously, the questions in the survey inquired students' perceptions of the theory-related and case-based videos' content, implementation quality and effectiveness of the videos in enhancing their PYD attributes [46, 47]. For the 15 theory-related videos and 9 case-based videos, all of the items received a high mean scores and positive response rates, showing that students had high satisfaction with the video content, quality of video delivery and the video effectiveness. In this focus group study, it was found that the majority of the students were satisfied with the video arrangement, content and design. Students also agreed that the videos could facilitate their pre-lesson self-learning, enhance their understanding of course content, and increase their participation in class discussion. Table 3 presents a side-by-side comparison of the findings from our first quantitative study and the findings from the present study. Both studies analyzed the same batch of students in Semester 2 of 2021/22 academic year. The comparison showed that the findings from the two studies were highly consistent. For example, in our quantitative evaluation study, one of the items "the animated video is well designed" received positive response rates from 94.69% to 98.20% across the 15 videos. Similarly, in our qualitative study, within the "video design" theme, the majority of the students thought that the quality of the videos (both theory-related and case-based videos) was good (N = 21). On the other hand, for the case-based videos, one of the items in the quantitative study "the video-based activity has increased my interaction with the teacher" received positive response rates from 86.32% to 96.39% across the nine videos. Comparably, in the current qualitative study, within the "videos' benefits to the students' class participation" theme, it was found that the students were able to answer teachers' questions after watching the videos (N = 16). These comparisons showed that the quantitative research findings supported the focus group study results, demonstrating that students had high satisfaction with and significant benefits received from watching the videos.

## 4.2. Limitations

Limitation of this focus group study should be noted. First, purposeful sampling was adopted in this study and a homogeneous group of participants was sampled. To further analyze the difference in perceptions between students based on their demographic variables, stratified purposeful sampling could be applied in future studies. Second, because of social distancing measures, the focus group interviews were conducted online, so physical features of group dynamics could not be observed. Richer analysis of group dynamics could be achieved if focus group interviews could be conducted in a face-to-face setting.

## 4.3. Implications for teaching practice

This study demonstrated the teaching and learning effectiveness of educational videos under a blended learning and flipped classroom arrangement. Education practitioners could consider piloting similar practice in their respective institutions in the future. These educational videos were very valuable under the COVID-19 pandemic [4, 113]. And they could be further used in other situations that require online education (e.g., courses for students in remote areas) [114, 115]. However, it should be noted that the effectiveness of educational videos may be different

**Table 3. Side-by-side comparison of quantitative and qualitative research findings.**

| Quantitative study | | Qualitative study | |
|---|---|---|---|
| **Questionnaire items for the 15 animated videos on theories/concepts** | **Positive response rates** | **Corresponding themes in the current focus group study** | **Corresponding findings** |
| 1. The animated video is well designed. | 98.20% to 94.69% | Video design | The majority of the students thought that the quality of the videos (both animated and case-based videos) was good (N = 21). |
| 2. The animated video is relevant to the lecture topic. | 98.91% to 94.62% | Video arrangement | In terms of the connection between the theory-related videos and the lectures, students believed that they were closely related because the teachers recapped the theories (N = 16), or discussed related personal experience and real-life examples during lectures (N = 21). |
| 3. The animated video is easy to understand. | 98.51% to 95.12% | Video design | The theory-related videos were very structured and explained the theories clearly. Model explanation was helpful to students' learning (N = 16). |
| 4. The animated video is interesting. | 96.65% to 90.81% | Video design | Students enjoyed the animation (N = 3), and the interesting animation helped their understanding (N = 10). |
| 5. The teacher used the animated video effectively. | 98.60% to 95.03% | Video arrangement | In terms of the connection between the theory-related videos and the lectures, students believed that they were closely related because the teachers recapped the theories (N = 16), or discussed related personal experience and real life examples during lectures (N = 21). |
| 6. The animated video has enabled me to understand related concepts. | 98.71% to 94.91% | Videos' benefits to the students' learning of course content | The theory-related videos contributed to students' learning through clear explanation of theories, which helped students to understand the content without prior knowledge. |
| 7. The animated video has deepened my understanding of the topic. | 98.43% to 94.42% | Videos' benefits to the students' learning of course content | The majority of the students thought that the videos (both animated and case-based videos) could facilitate their understanding of lecture content (N = 27) and related theories (N = 26). |
| 8. The animated video has enhanced my interests in the topic. | 96.64% to 90.62% | Videos' benefits to the students' learning of course content | The animation, layout and video cutting made the theory-related videos more interesting to students. |
| 9. The animated video has helped me to reflect on the topic. | 97.75% to 92.46% | Videos' benefits to the students' learning of course content | Some theory-related videos' content was related to students' daily lives, so they could stimulate students' self-reflection. |
| 10. I prefer more such animated video in the teaching and learning of this subject. | 97.57% to 94.11% | n/a | n/a |
| 11. The animated video can benefit my development. | 97.94% to 93.58% | Videos' benefits to the students' learning of course content | The skills introduced in the videos were useful to students. |
| 12. Overall speaking, I am satisfied with the animated video. | 98.52% to 94.42% | Videos' benefits to the students' learning of course content | The majority of the students rated the videos (both animated and case-based videos) 7 points (N = 10), 7 to 8 points (N = 7) or 8 points (N = 18) out of 10 points. |
| **Questionnaire items for the nine case-based videos** | **Positive response rates** | **Corresponding themes in this focus group study** | **Corresponding findings** |
| 1. The video-based activity is well designed. | 98.04% to 95.09% | Video design | The majority of the students thought that the quality of the videos (both animated and case-based videos) was good (N = 21). |
| 2. The video-based activity is relevant to the topic. | 98.97% to 97.74% | Video arrangement | In terms of the connection between the case-based videos and the lectures, the students thought that they were closely related because some teachers discussed the cases during lectures, while other teachers shared personal experience that were related to the videos (N = 20). |
| 3. The video is easy to understand. | 98.62% to 97.17% | Video content | The case-based videos were related to students' daily lives, so students could understand better and enjoy the videos (N = 7). |
| 4. The content of the video is interesting. | 97.02% to 92.48% | Video content | The case-based videos were related to students' daily lives, so students could understand better and enjoy the videos (N = 7). |
| 5. The content of the video is stimulating. | 98.02% to 96.05% | Video content | The case-based videos provided various perspectives and students could learn the ways they could solve similar problems in the future (N = 14). |
| 6. The video-based activity has increased my interaction with the teacher. | 96.39% to 86.32% | Videos' benefits to the students' class participation | After watching the videos, students were able to answer teachers' questions (N = 16). |
| 7. The video-based activity has increased my interaction with classmates. | 94.74% to 82.60% | Videos' benefits to the students' class participation | Many students perceived that both the theory-related videos and case-based videos could enhance their class participation (N = 16). |

(*Continued*)

**Table 3.** (Continued)

| Quantitative study | | Qualitative study | |
|---|---|---|---|
| Questionnaire items for the 15 animated videos on theories/concepts | Positive response rates | Corresponding themes in the current focus group study | Corresponding findings |
| 8. The teacher has used the video effectively. | 98.52% to 96.98% | Video arrangement | In terms of the connection between the case-based videos and the lectures, the students thought that they were closely related because some teachers discussed the cases during lectures, while other teachers shared personal experience that were related to the videos (N = 20). |
| 9. I have been very engaged in the video-based activity. | 97.87% to 92.61% | Videos' benefits to the students' class participation | The videos could also be used for class discussion, which increased students' motivation (N = 7). |
| 10. The video-based activity has deepened my understanding of the topic. | 98.32% to 96.87% | Videos' benefits to the students' learning of course content | The case-based videos applied theories to some real-life cases or social issue, so students could understand the concepts very easily. |
| 11. The video-based activity has enabled me to apply the theories in real life cases. | 97.67% to 93.60% | Videos' benefits to the students' learning of course content | The case-based videos were related to students' daily lives, so they could learn application of the theories in their daily lives. |
| 12. The video-based activity has enhanced my interests in the topic. | 97.21% to 92.84% | Videos' benefits to the students' class participation | Students were able to gain more interest in the lecture topics after knowing the content from the videos (N = 20). |
| 13. The video has helped me reflect on the subject matters. | 98.04% to 96.07% | Videos' benefits to the students' learning of course content | Students could have self-reflection after watching the case-based videos, which was helpful for them to gain more positive mindset. |
| 14. I prefer more such video-based activities in the teaching and learning of this subject. | 97.75% to 94.08% | n/a | n/a |
| Questionnaire Items on the Effectiveness of the Lecture | Positive Response rates | Corresponding themes in this focus group study | Corresponding Findings |
| 1. Overall speaking, the lecture has enriched my development. | 98.78% to 92.77% | n/a | n/a |
| 2. On the whole, I am very satisfied with this lecture. | 98.15% to 92.71% | n/a | n/a |

across disciplines [116]. Hence, education practitioners should adjust the video arrangement based on their situations and the needs of their students.

Moreover, this study demonstrated that various video design features could enhance university students' learning effectiveness, which supported the multimedia learning theories. The multimedia learning theories stated that the use of flow charts and tables and the highlighting of key points in multimedia reduced the extraneous cognitive load on learners, which could improve their learning efficiency [86, 89]. Also, the presentation of visual illustrations along with voiceover narrations could increase the learning efficiency of students [32, 90]. These findings not only supported the multimedia learning theories, but also provided design reference for education practitioners to develop university-level educational videos in the future. Interested parties could refer to the articles by Castillo et al. [117] and Brame [118] for comprehensive guidelines on the content design and the production of educational videos in a non-studio setting.

## 4.4. Recommendations for further research

The current research was conducted under the COVID-19 pandemic, when universities were forced to adopt synchronous or asynchronous online teaching to sustain teaching and learning amid the pandemic. Research participants in the current study needed to attend online synchronous lectures and watched the videos in a flipped classroom arrangement. Future research was recommended to be conducted in face-to-face or hybrid teaching context [119, 120], so the benefits of educational videos under different teaching and learning contexts could be examined. Moreover, the current study found that different video design features could enhance university students' learning effectiveness, supporting the multimedia learning theories. Because the application of multimedia learning theories in designing university-level

educational videos was relatively scarce when compared to studies in secondary education context [121, 122], further research should be conducted to strengthen our understanding on this topic.

Qualitative research employs a naturalistic approach to understand phenomenon in its natural or context-specific settings [53], so it allows researchers to examine deeper understanding of a phenomenon [123]. The current study demonstrated how the videos benefited students' pre-lesson self-learning and learning of course content, and enhanced their class participation. More qualitative research or mixed method studies could be implemented in the future, so we could accumulate evidence on how educational videos benefit students' learning, thus contributing to evidence-based teaching practice [124].

## 5. Conclusions

This study contributes to the literature by demonstrating students' positive perceptions of the video arrangement, design and content, and their subjective benefits from watching the 24 instructional videos in a leadership course covering psychosocial competence in a Hong Kong university. First, the study found that the pre-lesson video watching arrangement allowed students to have flexible self-learning, better understanding of lecture content, and more discussion during lectures. The pre-lesson blended learning exercise also engaged students and improved their understanding of lecture topics. Second, students perceived that various video features had contributed to their learning, including the pointers and lists in theory-related videos and the real-life filming in case-based videos. Third, in terms of video content, similarities between the content and students' daily lives or the social issues could improve students' learning effectiveness. Fourth, the clear explanation of theories in the theory-related videos and the case-based videos' down to earth content enabled students' to self-study the videos before lecture. Fifth, most students believed that the videos could facilitate their understanding of course content. Besides, students perceived that the two kinds of videos complement each other, with the theories explaining the cases and the cases demonstrating the application of theories. Sixth, students perceived that the videos could enhance their class participation because they could have more interest in the lecture topics and answer teachers' in-class questions after gaining understanding of the course content from the pre-lesson videos.

## Supporting information

**S1 Dataset.**
(DOCX)

## Author Contributions

**Conceptualization:** Daniel T. L. Shek, Lu Yu.

**Data curation:** Daniel T. L. Shek, Tingyin Wong, Xiang Li.

**Funding acquisition:** Daniel T. L. Shek.

**Methodology:** Daniel T. L. Shek.

**Project administration:** Daniel T. L. Shek, Xiang Li, Lu Yu.

**Supervision:** Daniel T. L. Shek, Xiang Li, Lu Yu.

**Writing – original draft:** Daniel T. L. Shek, Tingyin Wong.

**Writing – review & editing:** Daniel T. L. Shek, Xiang Li.

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
