## [Decision Letter · Decision Letter 0]

26 Jul 2023

PONE-D-23-18485Use of Instructional Videos in Leadership Education in Higher Education under COVID-19: A Qualitative StudyPLOS ONE

Dear Dr. Shek,

Thank you for submitting your manuscript to PLOS ONE. After careful consideration, we feel that it has merit but does not fully meet PLOS ONE’s publication criteria as it currently stands. Therefore, we invite you to submit a revised version of the manuscript that addresses the points raised during the review process.

Dear authors, Thank you for your submission to PLOS ONE. Please take a look at reviewer 1 comments and revise your paper accordingly.  Please highlight the changes and provide a point by point response to comments. Good Luck ==============================

We look forward to receiving your revised manuscript.

Kind regards,

Ehsan Namaziandost

Academic Editor

PLOS ONE

Journal Requirements:

"No competing interest"

Reviewers' comments:

Reviewer's Responses to Questions

**Comments to the Author**

1. Is the manuscript technically sound, and do the data support the conclusions?

Reviewer #1: Yes

Reviewer #2: Yes

2. Has the statistical analysis been performed appropriately and rigorously? 

Reviewer #1: N/A

Reviewer #2: Yes

3. Have the authors made all data underlying the findings in their manuscript fully available?

Reviewer #1: Yes

Reviewer #2: Yes

4. Is the manuscript presented in an intelligible fashion and written in standard English?

Reviewer #1: Yes

Reviewer #2: Yes

5. Review Comments to the Author

Reviewer #1: This study used a focus group method to assess the perceived quality and effectiveness of 24 instructional videos in a leadership course at a university. Data were collected during the year where students attended the lectures via an online synchronous teaching mode. Subjects were asked to watch two to three theory-related videos and scenario-based videos before attending the 2-hour online synchronous lecture such as flipped classroom approach in every week. Participants needed to complete the blended learning exercise and have reflections related to the videos before lecture each week. In total, 10 lectures that taught different leadership qualities, including intrapersonal development such as mental health and interpersonal development, used during video watching arrangement. A qualitative research method used via focus group interviews at the end of the semester conducting semi structured interviews. I believe this is an interesting study and it has a merit for this journal. Although authors mentioned triangulation at the end of the study, reliability and validity of measurements and results should be reported in detail including demographics of participants. In addition, more practical applications and future research suggestions should be added at the end of the manuscript. Thank you.

Reviewer #2: The Article was very helpful especially pre and post Covid era. The self study and self evaluation process was interesting to read. I did like the pre evaluation and the pre study approach I do suggest that a post evaluation will be helpful to determine the level of understanding of the topics presented. I also found that class participation of students after watching the videos was interesting and helpful to students. One point which I feel it is important is that self tests which I think should be given post lecture and the students should have the opportunity to repeat the tests multiple times in order to score the highest grade possible, and such evaluations for students should be in an interactive form of tests. Students currently are more prone to have an online self study class and this trend is increasing. This study was very interesting, I will only add different pre and post evaluation methods to measure the rate of success for students.

6. PLOS authors have the option to publish the peer review history of their article (what does this mean?). If published, this will include your full peer review and any attached files.

Reviewer #1: **Yes: **Ferman Konukman

Reviewer #2: **Yes: **SUDAD SAMAN

---

## [Author Response · Author response to Decision Letter 0]

26 Aug 2023

Please refer to the uploaded file.

---

## [Decision Letter · Decision Letter 1]

8 Sep 2023

Use of Instructional Videos in Leadership Education in Higher Education under COVID-19: A Qualitative Study

PONE-D-23-18485R1

Dear Dr. Shek,

We’re pleased to inform you that your manuscript has been judged scientifically suitable for publication and will be formally accepted for publication once it meets all outstanding technical requirements.

Kind regards,

Ehsan Namaziandost

Academic Editor

PLOS ONE

Additional Editor Comments (optional):

Reviewers' comments:

Reviewer's Responses to Questions

**Comments to the Author**

1. If the authors have adequately addressed your comments raised in a previous round of review and you feel that this manuscript is now acceptable for publication, you may indicate that here to bypass the “Comments to the Author” section, enter your conflict of interest statement in the “Confidential to Editor” section, and submit your "Accept" recommendation.

Reviewer #1: All comments have been addressed

Reviewer #2: All comments have been addressed

2. Is the manuscript technically sound, and do the data support the conclusions?

Reviewer #1: Yes

Reviewer #2: Yes

3. Has the statistical analysis been performed appropriately and rigorously? 

Reviewer #1: N/A

Reviewer #2: Yes

4. Have the authors made all data underlying the findings in their manuscript fully available?

Reviewer #1: Yes

Reviewer #2: Yes

5. Is the manuscript presented in an intelligible fashion and written in standard English?

Reviewer #1: Yes

Reviewer #2: Yes

6. Review Comments to the Author

Reviewer #1: Thank you very much for hard work and edits. All concerns answered. The paper has very good flow now. All sections organized very well. Practical applications provided more strength and validity and reliability of the study is very clear now. Best Regards.

Reviewer #2: I have reviewed the comments that was made in the original submission and did see the addition and the response to my comments. I do feel that the authors have addressed my concerns. I also have seen the author have made few changes to the language which makes the article be presented in a better English form. It also was helpful to see the data analysis . I wish the (N) will be bigger but this is not a concern to me.

7. PLOS authors have the option to publish the peer review history of their article (what does this mean?). If published, this will include your full peer review and any attached files.

Reviewer #1: **Yes: **Ferman Konukman

Reviewer #2: **Yes: **Sudad Saman

---

## [Editor Report · Acceptance letter]

12 Sep 2023

PONE-D-23-18485R1 

Use of Instructional Videos in Leadership Education in Higher Education under COVID-19: A Qualitative Study 

Dear Dr. Shek:

I'm pleased to inform you that your manuscript has been deemed suitable for publication in PLOS ONE. Congratulations! Your manuscript is now with our production department. 

Kind regards, 

on behalf of

Dr. Ehsan Namaziandost 

Academic Editor

PLOS ONE